# Management of Immunosuppression in Kidney Transplant Recipients Who Develop Malignancy

**DOI:** 10.3390/jcm8122189

**Published:** 2019-12-11

**Authors:** Danwen Yang, Natanong Thamcharoen, Francesca Cardarelli

**Affiliations:** 1Division of Nephrology, Beth Israel Deaconess Medical Center and Harvard Medical School, Boston, MA 02215, USA; yangdanwen@gmail.com (D.Y.); fcardare@bidmc.harvard.edu (F.C.); 2Section of Palliative Care, Division of General Medicine and Division of Nephrology, Beth Israel Deaconess Medical Center and Harvard Medical School, Boston, MA 02215, USA

**Keywords:** malignancy, cancer, kidney transplant, immunosuppression, graft failure, survival

## Abstract

The risk of cancer increases after transplantation. However, the consensus on immunosuppression (IS) adjustment after diagnosis of malignancy is lacking. Our study aims to assess the impact of IS adjustment on mortality of post-kidney transplant patients and allograft outcomes. We retrospectively reviewed the data in our center of 110 subjects. Our results showed IS dose adjustment was not statistically associated with mortality risk (HR 1.94, 95%CI 0.85–4.41, *p* = 0.12), and chemotherapy was the only factor that was significantly related to mortality (HR 2.3, 95%CI 1.21–4.35, *p* = 0.01). IS reduction was not statistically associated with worsening graft function (OR 3.8, 95%CI 0.77–18.71, *p* = 0.10), nor with graft survival (SHR 4.46, 95%CI 0.58–34.48, *p* = 0.15) after variables adjustment. Creatinine at cancer diagnosis and history of rejection were both negatively associated with graft survival (SHR 1.72, 95%CI 1.28–2.30, *p* < 0.01 and SHR 3.44, 95%CI 1.25–9.49, *p* = 0.02). Reduction of both mycophenolate and calcineurin inhibitors was associated with worsening graft function and lower graft survival in subgroup analysis (OR 6.14, 95%CI 1.14–33.15, *p* = 0.04; HR 17.97, 95%CI 1.81–178.78, *p* = 0.01). In summary, cancer causes high mortality and morbidity in kidney transplant recipients; the importance of cancer screening should be emphasized.

## 1. Introduction

The number of solid organ transplants has increased in the past decade, with 21,167 kidney transplants performed in the United States in 2018. Multiple studies have shown that there is an increased risk of malignancy in transplant recipients [1]. The overall cancer incidence rate is 90 per 1000 patients at 10 years after transplant, which is twice as high as in the general population, while the dialysis population has a 1.35 standardized cancer incident ratio compared to the general population [2]. Nonmelanoma skin cancer is even more frequent, with an incidence rate 14 times higher in transplant recipients compared to the general population. 

The burden of malignancy in kidney transplant patients is very high, and the mortality risk in kidney transplant recipients diagnosed with cancer is also greater than nontransplant patients. The median survival of kidney transplant patients with cancer is significantly lower than kidney transplant patients without cancer (2.1 years vs. 8.3 years). Malignancy is currently the second most common cause of death in kidney transplant patients after cardiovascular disease [3]. 

Despite the surging incidence of cancer in kidney transplant recipients, there is very limited data of how immunosuppression (IS) should be managed after malignancy diagnosis. In current practice, the consensus is that IS dose should be decreased in renal transplant patients with newly diagnosed malignancy, since there is evidence supporting that IS is associated with an increased risk of malignancy and can promote tumor growth [4]. However, specific recommendations regarding how to adjust IS after diagnosis of malignancy in kidney transplant patients are lacking, and management varies depending on institutions, and even by provider in the same practice. The regimen adjustment ranges from no dose reduction, dose reduction, or cessation of one or more immunosuppressive medications, to class switch. The aim of our study was to assess the impact of changes of IS on patient survival and graft function by retrospectively reviewing data on patients who were diagnosed with malignancy after kidney transplantation in our center. 

## 2. Methods

This is a retrospective data analysis, in which we identified subjects by manual search of medical records of patients who had kidney transplantations and cancer diagnosis from January 1990 to December 2018 at Beth Israel Deaconess Medical Center, Boston, MA, USA. Data on immunosuppressive regimen, creatinine at cancer diagnosis and one year after diagnosis were extracted from medical records. Time from transplant to cancer diagnosis, patient and graft survival data were calculated from actual dates. Study data were collected and managed using REDCap (Research Electronic Data Capture) electronic data capture tools hosted at Harvard Catalyst—Beth Israel Deaconess Medical Center [5]. Data were collected by chart review following HIPAA guidelines. Institutional Review Board approval was obtained for data collection and analysis, with a waiver for individual consent.

We included all adult patients (18 years or older) who were diagnosed with malignancy after renal transplantation, as seen in Figure 1. Nonmelanoma skin cancer patients who did not require chemotherapy or radiation for cancer treatment were excluded from the analysis. The primary outcome in this study was patient survival. Secondary outcome included graft failure (defined as renal replacement therapy requirement) and worsening renal function (defined as glomerular filtration rate (GFR) reduction of more than 30% or developed graft failure at one year after cancer diagnosis). The mortality and graft function information were obtained from medical records. Our primary variable of interest was dose reduction defined by any types of IS dose reduction. Variables considered to have potential confounding effect were included in the multivariable models, specifically we included demographics of the subjects (i.e., age, race, gender), creatinine at cancer diagnosis, history of rejection, cancer type, donor type, history of chemotherapy, and history of radiation therapy. Races were divided into black and nonblack, which includes Asian, Hispanic, and others. Cancer types were differentiated as solid organ malignancy and hematologic malignancy. Missing data and loss to follow-up were excluded from the analysis. For survival analysis, loss to follow-up cases were censored.

We stratified the population based on whether individuals had IS dose reduction. Means and standard deviations were used to summarize continuous variables with normal distribution. Median (interquartile range) was used for skewed continuous variables. Categorical variables were summarized as percentage. We used t-test to assess the differences in continuous variables that were normally distributed. We tested the difference in categorical variables with Fisher’s exact test. Wilcoxon rank sum test was used to test skewed continuous variables. We used logistic regression to assess variables for worsening graft function at one year after cancer diagnosis. For graft failure outcome, competing risk survival analysis (Fine and Gray model) was used to assess cumulative graft failure incidence, and the covariable effect on graft failure was reported as subdistribution hazard ratio. Death without graft failure was considered as a competing outcome. The Cox proportional hazards model was used to assess risk factors for mortality, and the data was censored by last follow-up date. Patient survival was analyzed using the Kaplan–Meier method with significance tested using the log-rank test. 

For subgroup analysis, patients were divided into groups according to type of IS reduction (mycophenolate mofetil (MMF), calcineurin inhibitor (CNI), and reduction of both MMF and CNI). We compared each group to the group without any IS changes to assess the risk of worsening graft function and graft failure between these groups. Propensity score adjustment was utilized for subgroup analysis given the small number of subjects in each group. Propensity score of each subject was calculated based on significant factors derived from initial analysis of worsening graft function and graft failure. Then, we performed regression analysis for worsening graft function outcome and Cox regression model for graft failure outcome. Propensity score was applied to the model for adjustment.

All multivariable models ware built based on clinical risk factors and statistically significant variables from univariable analyses. *p* < 0.05 was considered to be significant. The data collected were analyzed using the Stata software version 15.0 (Stata Corp., College Station, TX, USA).

## 3. Results

### 3.1. Baseline Characteristics of the Study Population

One hundred and ten subjects who underwent kidney transplantation and developed malignancy were included in our analysis, as seen in Figure 1. Patients’ demographics are shown in Table 1. 

The mean age at cancer diagnosis was 60.2 years. Male gender contributed to 65.5% of subjects. The ethnicities of subjects were 77.3% non-Hispanic White, and 11.8% non-Hispanic Black. Our study population underwent transplantation during 1971–2018 (1971–1999 in 24 patients and 2000–2019 in 86 patients). Of the study population, 73.6% underwent IS regimen changes (dose reduction or class switch), 26.4% patients had no changes in their IS regimen. Among patients with IS reduction, 26 patients had reduction of both mycophenolate mofetil (MMF) and calcineurin inhibitor (CNI), 19 patients had reduction of CNI only, while 25 patients had reduction of MMF only. 

The IS regimen of our patient population is presented in Figure 2. Degree of dose reduction for each IS was showed as median of percent reduced from precancer diagnosis dose in Table 2. 

Medians of percent dose reduction were 60% for tacrolimus, 100% (completely discontinued) for cyclosporine, and MMF or mycophenolic acid. Solid organ malignancies represented 79.1% of the cases; the remainders were hematological cancers. Number of subjects for each type of malignancy are shown in Figure 3.

Deceased donor kidney transplant constituted 51.8% of the transplants, and the remainders were from living donors. Mean baseline creatinine at time of cancer diagnosis was 1.6 mg/dL (interquartile range 1.1–1.8 mg/dL). Median time of cancer diagnosis was 6.76 years after transplantation (interquartile range 2.7–11.7 years).

### 3.2. Mortality

The mortality rate was very high, at 46.4 % (51/110), with median survival time of 1.8 years after cancer diagnosis (interquartile range 0.7–5.6 years). Thirty patients died within one year of cancer diagnosis. Analysis of mortality in the transplantation eras before and after 2000 was performed by chi-square test, mortality rate between both eras was not statistically significant, *p* = 0.65. Of 51 patients who died, malignancy was the cause of death in 27 patients. Infection was the cause of death in four patients. Eighteen patients had no cause of death recorded. Other causes of death were cardiovascular disease and unknown cause. Kaplan–Meier curve and log-rank test revealed that IS dose reduction significantly increased mortality, *p* = 0.01, as seen in Figure 4. 

We performed univariate Cox regression analysis to assess relationship of each variable to mortality, as shown in Table 3. According to our univariate regression analysis model, older age, male gender, IS dose reduction, and chemotherapy were associated with higher mortality. However, in the multivariate model, only chemotherapy remained significant (HR 2.3, 95%CI 1.21–4.35, *p* = 0.01). When we excluded patients who died within six months of cancer diagnosis, the results did not change. 

We also checked the interaction between chemotherapy and dose reduction; the *p* value of 0.36, indicates no strong interaction between those two variables. The spearman correlation coefficient between chemotherapy and dose reduction was 0.28.

### 3.3. Worsening Graft Function

There were 100 patients who had post-cancer diagnosis creatinine at one year available. Twenty percent of patients (20/100) developed worsening graft function. In univariate logistic regression, creatinine at cancer diagnosis and female gender were associated with worsening renal function. Those variables remained significant in the multivariable analysis after adjusting for creatinine at cancer diagnosis, IS dose reduction, age, and gender. Interestingly, cancer type, chemotherapy, and donor type were not associated with worsening graft function at one year. The result is shown in Table 4.

It is important to note that the direction and magnitude of the estimates for IS dose reduction suggest a potentially strong effect on worsening graft function and mortality outcome, but our study did not have enough power to detect this, given the small number of patients.

### 3.4. Graft Failure

In our study, the graft failure rate was 16.4% (18/110). Median graft survival after cancer diagnosis in patients with graft failure was 2.97 years (interquartile range 0.56–4.22 years). Causes of graft failure were acute kidney injury in five patients, “chronic allograft nephropathy” in five patients, and acute rejection in five patients. BK nephropathy, multiple myeloma, and unknown cause contributed to the remaining patients. 

As shown in Table 5, in competing risk survival model, creatinine at cancer diagnosis, history of rejection and hematologic cancer were associated with increased risk of graft failure in univariable analysis. After adjusting for age at cancer diagnosis, creatinine at cancer diagnosis, IS dose reduction, malignancy type, and history of rejection, our result showed that creatinine at cancer diagnosis and history of rejection have remained statistically significant with SHR 1.72, 95% CI 1.28–2.30, *p* < 0.01 and SHR 3.44, 95% CI 1.25–9.49, *p* = 0.02, respectively. 

IS was reduced in all the patients who had graft failure, except for one patient who did not have his IS adjusted, as he was only on low dose tacrolimus monotherapy due to BK viremia. PTLD diagnosis contributed to five out of 18 cases of graft failure.

### 3.5. Subgroup Analysis 

#### 3.5.1. Worsening Graft Function

We performed subgroup analysis in patients who had IS reduction, defined by reduction of CNI (19 patients), reduction of MMF (25 patients), and reduction of both (29 patients), compared to 29 patients who had no IS change at all to analyze their impact on worsening graft function at one year. After adjusting for gender, age at cancer diagnosis, creatinine at cancer diagnosis using propensity score, reduction of two types of IS was a significant factor for worsening graft function at one year in logistic regression, OR 6.14, 95% CI 1.14–33.15, *p* = 0.04, as seen in Table 6.

#### 3.5.2. Graft Failure

Subgroup analysis was also performed to assess the impact of different IS reduction regimens on graft failure. The patient groups are the same as subgroup analysis in worsening graft function. In the Cox model adjusted for age at cancer diagnosis, creatinine at cancer diagnosis, history of rejection, and cancer type using propensity score, reduction of both CNI and MMF was associated with graft failure, HR 17.97, 95%CI 1.81–178.78, *p* = 0.01, as seen in Table 7.

## 4. Discussion

Although there is increasing evidence of high morbidity and mortality of kidney transplant patients diagnosed with malignancy, specific recommendation on how to adjust IS is lacking. A randomized trial comparing low cyclosporine dose to regular dose found no difference in graft survival or function, although the low-dose regimen was associated with fewer malignant disorders and more frequent rejections [6]. Another randomized controlled trial in 489 kidney transplant patients with 20-year follow-up showed that azathioprine and cyclosporine-based regimens were associated with similar overall long-term cancer risks. In addition, gender, previous antithymocyte globulin (ATG) exposure, and graft failure showed no association with development of malignancy, excluding skin cell carcinoma [7]. One retrospective observational study in heart transplant patients showed that everolimus treatment was associated with lower malignancy risk than MMF [8]. Previous studies showed that sirolimus was associated with reduction in the risk of malignancy and nonmelanoma skin cancer in kidney transplant recipients; however, it was associated with increased mortality risk [9].

KDIGO guidelines published in 2010 recommend considering a reduction of IS for kidney transplant recipients with malignancy (2C recommendation). Important factors to consider (not graded) include the stage of cancer at diagnosis, malignancies which are likely to be exacerbated by IS, available therapies, and whether IS interferes with ability to administer standard chemotherapy [10]. The likelihood of cancer being exacerbated by IS can be assessed using standardized incidence ratio (SIR), which compares the malignancy risk in kidney transplant patients to that in the general population. Cancers with SIR > 3, such as Kaposi’s sarcoma, PTLD, and ano-genital cancer, are mostly associated with viral infections, e.g., Human Herpesvirus 8 (HHV8), Epstein–Barr virus (EBV), human papillomavirus (HPV). It has been shown that the incidence of Kaposi’s sarcoma, non-Hodgkin’s lymphoma, HPV related ano-genital cancer, and melanoma were significantly elevated in patients with functioning transplant graft, but not after transplant failure, when patients were back on dialysis, suggesting that IS has significant effect on these types of cancer. As a consequence, IS adjustment should be strongly considered in these types of malignancy [11,12].

Our study showed that mortality rate in kidney transplant patients with diagnosis of malignancy was high (46.4%), with median survival time of 1.8 years after cancer diagnosis (interquartile range 0.7–5.6 years). Mortality rate was not significantly different between patients who had transplantation before and after year 2000. Interestingly, in our study, malignancy was the main cause of death in subjects whose cause of death was recorded, while the leading cause of death in kidney transplant recipients in general is cardiovascular disease. This data suggests that malignancy contributes to major of mortality in kidney transplant recipients with cancer diagnosis. In addition, more than half of deceased subjects died within two years of their cancer diagnosis, possibly reflecting advanced cancer at presentation and/or aggressive disease in transplant patients. Our data emphasizes that the appropriate cancer screening could reduce mortality and its importance should be particularly stressed in transplant recipients.

The possible causes of increased mortality risk in this population have been attributed to reduction of immune surveillance in the setting of IS and limited use of certain chemotherapy regimens due to reduced renal function. Notably, kidney transplant recipients and patients with HIV share a similar pattern of increased risk of cancer. Consequently, the increased risk of malignancy after kidney transplantation is thought to be caused by viral infection along with chronic IS use [2].

The significant variable between dose reduction and no reduction groups was whether patients required chemotherapy, suggesting that physicians are more inclined to reduce IS when the cancer is more advanced. The type of cancer (hematologic or solid organ malignancy) did not appear to affect the decision of changing the IS. According to Kaplan–Meier analysis, mortality was significantly higher in the dose reduction group, which is likely confounded by the fact that patients with more advanced stage malignancy tended to have their IS adjusted. Our result is comparable to a previous study in a different center [13]. For multivariate analysis, our study demonstrated that chemotherapy is the only variable associated with mortality, which could be similarly explained by the severity of disease.

As expected, patients with baseline poor kidney function had higher risk of graft failure. The degree of IS dose reduction was significant in majority of patients (IS dose was reduced by at least 50% to completely stopped) putting patients at higher risk of acute allograft rejection. Interestingly, our data showed a novel and important factor in subgroup analysis, reduction of both CNI and MMF put patients at higher risk of graft failure. As a consequence, we recommend that providers should carefully weigh the risks and benefits before drastically changing IS in transplant recipients after cancer diagnosis. A multidisciplinary approach is necessary, focusing on the individual patient’s wishes and goals in terms of survival, quality of life, and factor in the possibility of graft failure and return to dialysis. Patients with renal allograft failure returning to dialysis seem to have inferior quality of life and higher rate of depression compared to wait-listed transplant naive patients [14].

Based on our cohort, patients with PTLD had the highest mortality (seven out of 17 patients). Graft failure incidence in patients diagnosed with PTLD was also the highest compared to any other malignancy, as five out of 18 patients who had graft failure were diagnosed with PTLD.

Our study has many limitations. First, it is an uncontrolled retrospective study; therefore, the direct and independent effect of IS changes on mortality could not be clearly determined. Second, our database is from a single center, which has a relatively small number of subjects and heterogeneous cancer types, which might contribute a major confounder. Third, despite adjusting for chemotherapy and radiation therapy, cancer staging was not included in our analysis due to lack of record and heterogeneity of cancer diagnosis. While some chemotherapy regimens could have been a cause graft failure, we did not include this data in our analysis. Lastly, we disregarded the effect of sirolimus and steroid adjustment since both drugs are not part of the standard immunosuppressive regimen at our transplant center.

## 5. Conclusions

Our study shows no difference in mortality and graft survival outcomes between reduction and no reduction of IS in kidney transplant recipients diagnosed with cancer. However, it is important to note that the direction and magnitude of the estimates for IS dose reduction suggest a potentially strong effect on worsening graft function and mortality outcome, but a lacking power, caused by the small group of subjects, prevented us to detect the differences. The mortality rate in this population is high and malignancy is usually aggressive; therefore, kidney transplant patients would benefit from early detection of disease by routine cancer screening. The data from our study reveals a novel finding: the risk of graft failure appears remarkably higher after adjusting two immunosuppressive medications. Most importantly, providers should have an extensive discussion with patients regarding the risk and benefit of IS adjustment, chances of prolonging survival from cancer treatment, and worsening quality of life in case patients develop kidney allograft failure requiring dialysis. As a future direction, a prospective study might be the key to define the temporal effect of IS adjustment on patient’s survival, malignancy, and allograft outcomes in kidney transplant recipients.

## Figures and Tables

**Figure 1 jcm-08-02189-f001:**
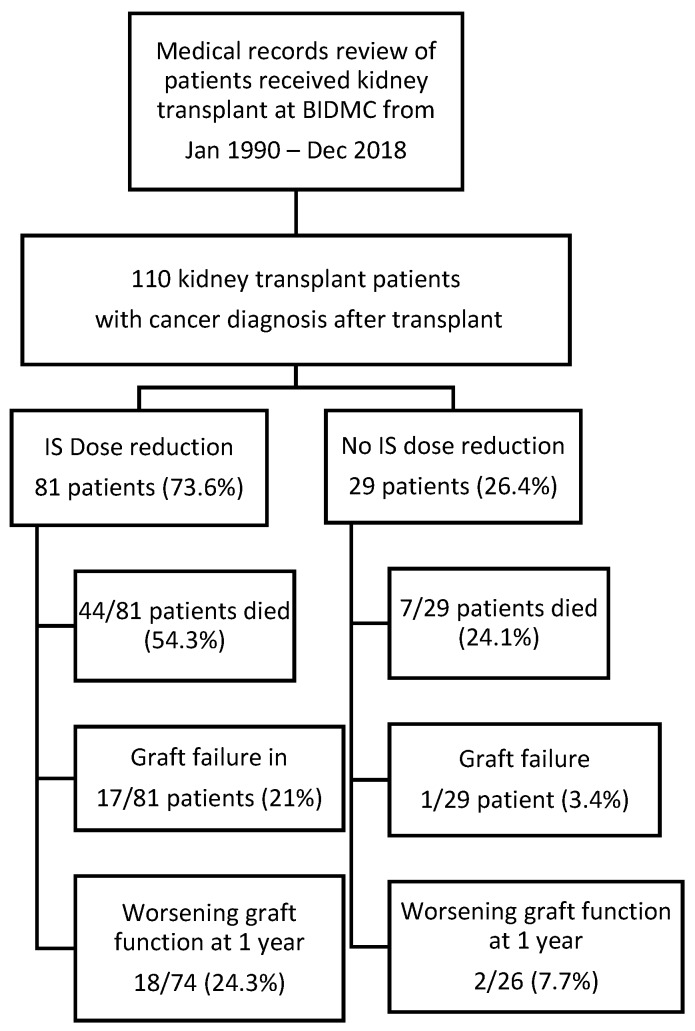
Summary of the study.

**Figure 2 jcm-08-02189-f002:**
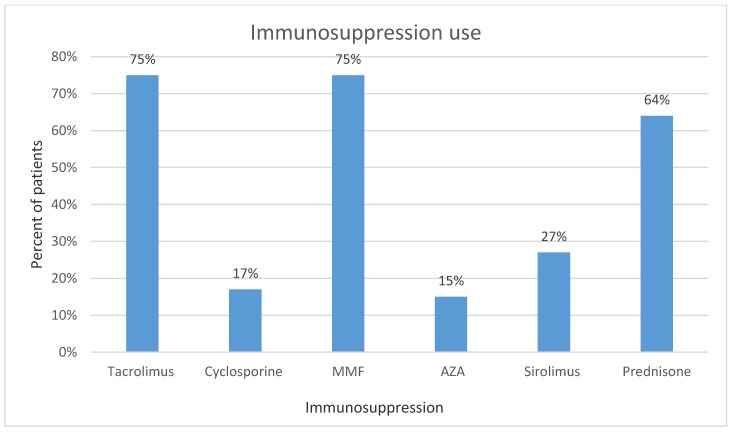
Types of immunosuppression (IS) used by subjects in the study. MMF = mycophenolate mofetil, AZA = azathioprine.

**Figure 3 jcm-08-02189-f003:**
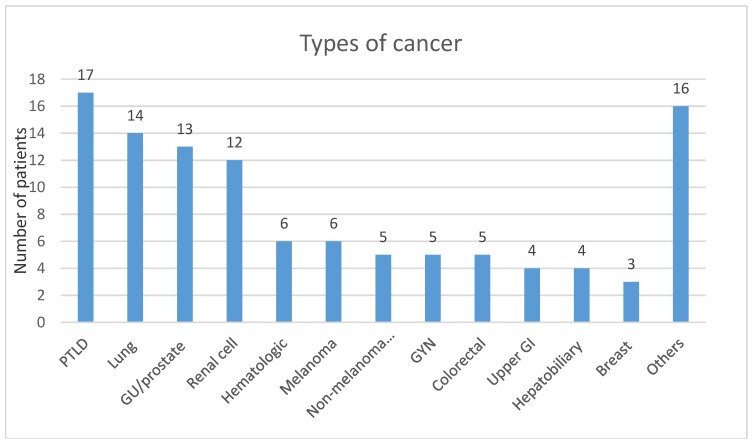
Number of patients in each type of cancer; PTLD = Post-transplant Lymphoproliferative Disorders, GU = Genitourinary, GYN = gynecology, GI = gastrointestinal. Other cancers are head/neck, Kaposi sarcoma, other sarcoma, brain, and unknown origin.

**Figure 4 jcm-08-02189-f004:**
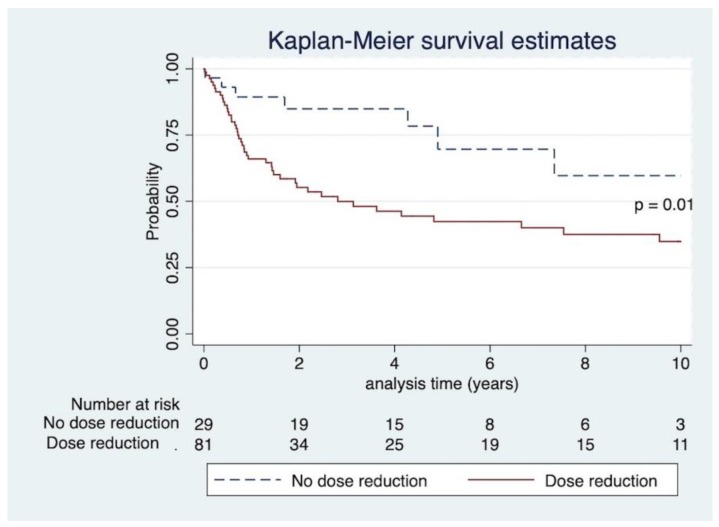
Kaplan–Meier curve and log-rank test of IS dose management and mortality risk.

**Table 1 jcm-08-02189-t001:** Baseline characteristic of subjects (N = 110).

Characteristics	Dose Reduction (N = 81)	No Dose Reduction (N = 29)	*p* Value
Sex			
Male	55 (67.9%)	17 (58.6%)	0.37
Race			
White	61(75.3%)	24 (82.8%)	0.17
Black	8 (9.9%)	5 (17.2%)	
Asian	6 (7.4%)	0	
Hispanic	6 (7.4%)	0	
Age at cancer diagnosis	60.1 (11.2)	60.2 (9.1)	0.87
Primary disease			
Diabetes	28 (34.6%)	7 (24.1%)	0.33
Glomerulonephritis	15 (18.5%)	8 (27.6%)	
PKD	7 (8.6%)	3 (10.3%)	
Reflux	1 (1.2%)	2 (6.9%)	
Other	30 (37.0%)	9 (30.0%)	
Transplant type			
Deceased donor	41 (50.6%)	16 (55.2%)	0.77
Living unrelated donor	24 (29.6%)	9 (31.0%)	
Living related donor	16 (19.8%)	4 (13.8%)	
Mean creatinine at cancer diagnosis (mg/dL)	1.65	1.49	0.39
	(1.44–1.87)	(1.27–1.70)	
Type of cancer			
Hematological cancer	19 (23.5%)	4 (13.8%)	0.27
Solid organ cancer	62 (76.5%)	25 (86.2%)	
History of chemotherapy	48 (59.3%)	8 (27.6%)	<0.01
History of radiation	27 (33.3%)	11 (37.9%)	0.66
History of rejection	14 (17.3%)	5 (18.5%)	0.22

**Table 2 jcm-08-02189-t002:** Median percent dose reduction of each immunosuppression (100% = completely discontinuation of immunosuppression).

Immunosuppression	Median Percent Dose Reduction (IQR)
Tacrolimus	60% (29.17%–100%)
Cyclosporine	100% (100%–100%)
Mycophenolate mofetil (MMF) or mycophenolic acid	50% (50%–100%)

**Table 3 jcm-08-02189-t003:** Effect of immunosuppression dose reduction on patients’ mortality. Multivariable analysis was adjusted for age, IS dose reduction, chemotherapy history, and gender. Nonblack race = White, Asian, Hispanic, and other races. * = Statistically significant, *p* < 0.05.

Variables	Univariate Model		Multivariable Model	
	Hazard Ratio (95%CI)	*p*-Value	Hazard Ratio (95%CI)	*p*-Value
Age at cancer diagnosis	1.04 (1.02–1.07)	<0.01 *	1.02(0.99–1.05)	0.13
IS dose reduction	2.68 (1.21–6.00)	0.02 *	1.94 (0.85–4.41)	0.12
Chemotherapy	3.08 (1.69–5.61)	<0.01 *	2.30 (1.21–4.35)	0.01 *
Male	2.44 (1.24–4.77)	0.01 *	1.97 (0.98–3.99)	0.06
History of rejection	0.78 (0.37–1.67)	0.53		
Cr at cancer diagnosis	1.06 (0.80–1.42)	0.68		
Black Race ^+^	0.36 (0.11–1.17)	0.09		
Solid organ cancer	1.24 (0.58–2.65)	0.57		
Radiation therapy	1.59 (0.91–2.79)	0.10		
Deceased donor	1.62 (0.92–2.84)	0.09		

**Table 4 jcm-08-02189-t004:** Impact of immunosuppression dose reduction on worsening GFR > 30% at one year after cancer diagnosis. Multivariable analysis was adjusted for age, creatinine at cancer diagnosis, IS dose reduction, and gender. Nonblack race = White, Asian, Hispanic, and other race. * = Statistically significant, *p* < 0.05.

Variables	Univariable Model		Multivariable Model	
	Odds Ratio (95%CI)	*p*-Value	Odds Ratio (95%CI)	*p*-Value
Age at cancer diagnosis	0.99 (0.94–1.03)	0.59	1.02 (0.97–1.08)	0.48
Cr at cancer diagnosis	2.37 (1.28–4.40)	<0.01 *	2.67 (1.35–5.28)	<0.01 *
IS dose reduction	3.86 (0.83–17.94)	0.09	3.80 (0.77–18.71)	0.10
Male	0.43 (0.16–1.16)	0.01 *	0.22 (0.06–0.77)	0.02 *
Black Race ^+^	0.33 (0.04–2.72)	0.30		
Solid organ cancer	0.54 (0.18–1.63)	0.27		
Chemotherapy	2.05 (0.74–5.68)	0.17		
Radiation therapy	0.47 (0.13–1.36)	0.15		
Deceased donor	1.11 (0.41–2.96)	0.84		

**Table 5 jcm-08-02189-t005:** Impact of immunosuppression dose reduction on graft survival. Multivariable analysis was adjusted for age, creatinine at cancer diagnosis, history of rejection, IS dose reduction, and cancer type. Nonblack race = White, Asian, Hispanic, and other races. * = Statistically significant, *p* < 0.05.

Variables	Univariable Model		Multivariable Model	
	SHR (95%CI)	*p*-Value	SHR (95%CI)	*p*-Value
Age at cancer diagnosis	0.97 (0.93–1.01)	0.16	0.99 (0.94–1.03)	0.62
Cr at cancer diagnosis	1.83 (1.45–2.30)	<0.01 *	1.72 (1.28–2.30)	<0.01 *
History of rejection	3.63 (1.45–9.08)	0.01 *	3.44 (1.25–9.49)	0.02 *
IS dose reduction	6.19 (0.82–46.73)	0.08	4.46 (0.58–34.48)	0.15
Solid organ cancer	0.35 (0.13–0.95)	0.04 *	0.48 (0.16–1.42)	0.18
Black Race ^+^	0.91 (0.23–3.61)	0.90		
Male	0.67 (0.27–1.66)	0.39		
Chemotherapy	1.39 (0.56–3.46)	0.48		
Radiation therapy	0.93 (0.36–2.44)	0.89		
Deceased donor	0.80 (0.32–1.98)	0.62		

**Table 6 jcm-08-02189-t006:** Impact of each type of IS reduction compared to no dose reduction on worsening GFR > 30% at one year after cancer diagnosis Adjusted for gender, age at cancer diagnosis, and creatinine at cancer diagnosis. * = Statistically significant, *p* < 0.05.

Immunosuppression Reduction (N)	OR (95% CI)	*p*-Value
CNI Reduction (19/29)	1.31 (0.16–10.59)	0.80
MMF Reduction (25/29)	5.28 (0.86–32.55)	0.07
Reduction of all IS (26/29)	6.14 (1.14–33.15)	0.04 *

**Table 7 jcm-08-02189-t007:** Impact of each type of IS reduction compared to no dose reduction on graft survival. Adjusted for age at cancer diagnosis, creatinine at cancer diagnosis, history of rejection and cancer type. * = Statistically significant, *p* < 0.05.

Immunosuppression Reduction	HR (95%CI)	*p*-Value
CNI Reduction (19/29)	6.52 (0.46–92.70)	0.17
MMF/myfortic Reduction (25/29)	0.66(0.04–11.14)	0.77
Reduction of all IS (26/29)	17.97 (1.81–178.78)	0.01 *

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
