# Peer review of "Management of Immunosuppression in Kidney Transplant Recipients Who Develop Malignancy"

_jcm, 2019, doi:10.3390/jcm8122189_

Round 1
Reviewer 1 Report
Reviewer Comments
This manuscript discussed the management of immunosuppression (IS) in kidney transplant recipients who develop malignancy. Post-transplantation the risk of cancer increases and authors highlight the impact of IS adjustment on post-transplantation mortality and graft function. The authors emphasized that cancer is the second leading cause of death in kidney transplant patients after cardiovascular diseases. The authors have done extensive study to provide adequate conclusions but there are some concerns that must be addressed. So, the authors should consider the following comments for minor improvement prior to re-submission.
1 Describe the clinical design conducted in detail. A flow-chart summarizing the study would be ideal.
2 Deceased donor kidney transplant constituted 51.8% of the transplants, and the remainders were from living donors. Is there any difference between kidney transplant from deceased and live donors in terms creatinine ratio and graft rejection?
3 Table 4: A different form of representation Pie-chart or Bar-graph may enhance the representation of data.
4 Any difference statistically between males and females on cancer diagnosis, immunosuppression dose reduction, and graft rejection or function?
5 One hundred patients, who had creatinine at 1 year after cancer diagnosis recorded, were analyzed for this outcome. Worsening graft function was defined as GFR reduction of more than 30 % or requiring renal replacement therapy at 1 year after cancer diagnosis . What was the baseline creatinine ratio at the time of diagnosis?
6 Worsening graft function was defined as GFR reduction of more than 30 or requiring renal replacement therapy at 1 year after cancer diagnosis. The statement is confusing as the authors previously mentioned that after post-transplantation the risk of cancer increases. Is the RRT being performed here on kidney transplant patients? Needs to be clear in the text.
7 After adjusting for gender, age at cancer diagnosis, creatinine at cancer diagnosis using propensity score, reduction of two types of IS was a significant factor for worsening graft function at one year in logistic regression, OR 6.14 95% CI 1.14-33.15, p =0.04 (table 8). How was the adjustment done? Explain briefly in methods section.
8 Table 8: Is reduction of MMF and all IS comparable?
9 Was there any independent effect of IS on mortality or graft failure?
10 The data from our study reveals a novel finding, risk of graft failure appears remarkably higher after adjusting 2 immunosuppressive medications. Based on the above finding, what IS dosage or IS combination would authors recommend for increasing quality of life in kidney transplant patients?
11 The conclusion should include a future direction to further advance the study.
Author Response
Dear reviewers,
We would like to thank you for very thoughtful comments. We have revised the manuscript accordingly. Please find attached a point-by-point response to reviewer’s concerns.
1 Describe the clinical design conducted in detail. A flow-chart summarizing the study would be ideal.
Flow chart was added to the manuscript, Figure1.2 Deceased donor kidney transplant constituted 51.8% of the transplants, and the remainders were from living donors. Is there any difference between kidney transplant from deceased and live donors in terms creatinine ratio and graft rejection?
Kidney graft type (deceased vs live) is one of variable in our model. According our result in table 5 and 6, there is no difference of worsening creatinine and graft failure between deceased donor and live donor group. We did not use graft rejection as an outcome since the number of patients with graft rejection was very small. We used graft failure as an outcome instead.3 Table 4: A different form of representation Pie-chart or Bar-graph may enhance the representation of data.
The table was changed to Bar graph as recommended.4 Any difference statistically between males and females on cancer diagnosis, immunosuppression dose reduction, and graft rejection or function?
Difference of sex on cancer diagnosis: We do not have this data. We only included patient with cancer diagnosis in the study, therefore we did not have the total number of patients who received transplant. We did not calculate the ratio of patient who developed cancer for each sex. Difference of sex on immunosuppression dose reduction: Not statistically difference, showed in table 1. Difference of sex on graft rejection or function: We included sex/gender as one of variables in all models in all analyses. Table 5 showed male as a risk factor for higher mortality. Table 6 showed male has lower risk of worsening graft function at one year. Sex is not a significant factor for graft survival according to Table 7.5 One hundred patients, who had creatinine at 1 year after cancer diagnosis recorded, were analyzed for this outcome. Worsening graft function was defined as GFR reduction of more than 30 % or requiring renal replacement therapy at 1 year after cancer diagnosis. What was the baseline creatinine ratio at the time of diagnosis?
Baseline creatinine was showed in Table 1, patients’ demographic/characteristics data.6 Worsening graft function was defined as GFR reduction of more than 30 or requiring renal replacement therapy at 1 year after cancer diagnosis. The statement is confusing as the authors previously mentioned that after post-transplantation the risk of cancer increases. Is the RRT being performed here on kidney transplant patients? Needs to be clear in the text.
Worsening graft function is defined as patient had reduction in GFR > 30 at 1 year after cancer diagnosis or required RRT at 1 year after cancer diagnosis. We only included patients who had functional graft/did not require RRT at baseline (at cancer diagnosis).7 After adjusting for gender, age at cancer diagnosis, creatinine at cancer diagnosis using propensity score, reduction of two types of IS was a significant factor for worsening graft function at one year in logistic regression, OR 6.14 95% CI 1.14-33.15, p =0.04 (table 8). How was the adjustment done? Explain briefly in methods section.
Added the explanation in method section.8 Table 8: Is reduction of MMF and all IS comparable?
In this analysis, we disregarded the amount of IS dose that was reduced. However, the median of percent dose reduction was showed in table 3.9 Was there any independent effect of IS on mortality or graft failure?
Yes, IS reduction is a statistically significant factor for mortality from univariable analysis showed in table 5, but not graft failure (table 7)10 The data from our study reveals a novel finding, risk of graft failure appears remarkably higher after adjusting 2 immunosuppressive medications. Based on the above finding, what IS dosage or IS combination would authors recommend for increasing quality of life in kidney transplant patients?
We could not specify which regimen or how much we should decrease IS dose to avoid graft failure since we only have small groups of subjects.11 The conclusion should include a future direction to further advance the study.
The future direction is already mentioned, line 304. (A prospective study should be performed to define the temporal effect of IS adjustment on patient’s survival, malignancy and allograft outcomes in kidney transplant recipients.)We hope that you find our responses satisfactory and that the manuscript is now acceptable for publication.
Best Regards,
Natanong Thamcharoen, MD
Reviewer 2 Report
The authors performed a well-balanced analysis and the study provides an important information about the use of immunosuppressive medications or adjustments post kidney transplant. The data support the conclusion and the study would be clinically beneficial.
In the current manuscript, Yang et al studied the management of immunosuppression in kidney transplant recipients who developed malignancy. The authors highlighted the impact of immunosuppression (IS) adjustment on mortality of post-kidney transplant patients and allograft outcomes. The main finding of the study is the mortality of patients after kidney transplant who developed with cancer does not improve after adjusting IS, however, it actually worsens graft function and survival. Further, the authors concluded cancers and chemotherapy mainly causes the high mortality and morbidity in kidney transplant recipients. The manuscript is well studied and a few minor corrections below will help to improve the quality of the manuscript.
Minor comments:
Line 38, a sentence describing mortality risk in kidney transplant recipients who has pre and post history of diagnosed cancer would help the readers to compare the risk factor of pre and post diagnosed cancer in kidney transplant and allograft outcomes
Line 53, methods, provide a flow chart that summarizes the clinical design and data analysis
Line 66, the author mentioned’ we included all adult patients’, do they include both males and females, if yes please specify numbers and how they are categorized.
Table 2, 3, 4, 8 and 9, providing a simple representative bar diagram is more easy and effective for the readers.
Line 156, the sentence ‘One hundred patients, who had creatinine at 1 year after cancer diagnosis recorded’ is confusing. Please re-write this sentence.
Line 160, the author states that’ creatinine at cancer diagnosis and female gender were associated with worsening renal function’ however no data shown in the table 6.
The authors have sufficient references to support their data and the text are readable.
Author Response
Dear Reviewer,
We would like to thank you for very thoughtful comments to our manuscript. We revised the manuscript accordingly. Point-by-point response to the comments is below.
Line 38, a sentence describing mortality risk in kidney transplant recipients who has pre and post history of diagnosed cancer would help the readers to compare the risk factor of pre and post diagnosed cancer in kidney transplant and allograft outcomes
- The sentence was added to revised manuscript.
Line 53, methods, provide a flow chart that summarizes the clinical design and data analysis
- Added to the revised manuscript (Figure 1)
Line 66, the author mentioned’ we included all adult patients’, do they include both males and females, if yes please specify numbers and how they are categorized.
- Subjects' demographic data including gender is shown in table 1(result section).
Table 2, 3, 4, 8 and 9, providing a simple representative bar diagram is more easy and effective for the readers.
- Changed table 2 and 4 to bar diagram as shown in revised manuscript. Table 3 was not changed. Table 8 and 9 demonstrate odd ratio, we think it is appropriate to be presented as they are.
Line 156, the sentence ‘One hundred patients, who had creatinine at 1 year after cancer diagnosis recorded’ is confusing. Please re-write this sentence.
- The sentence has been revised.
Line 160, the author states that’ creatinine at cancer diagnosis and female gender were associated with worsening renal function’ however no data shown in the table 6.
- According to table 6, OR of creatinine at cancer diagnosis variable are 2.37(univariate model) and 2.67(multivariable model) with significant p value, thus patient with higher creatinine at cancer diagnosis is at risk of worsening graft function at 1 year post diagnosis. Regarding gender variable, male sex is the reference group. OR for male sex factor < 1 implies that OR would be invert (> 1) if female sex was used as the reference group. Therefore, we concluded that female sex is a risk factor for worsening graft outcome.
We hope you would find the revision satisfactory.
Regards,
Natanong Thamcharoen, MD